# Author Name Disambiguation via Paper Association Refinement and Compositional Contrastive Embedding

## ABSTRACT

Author name disambiguation (AND) is an essential task for online academic retrieval systems. Recent models adopt representation learning in the author's name disambiguation. Despite achieving remarkable success, these methods may be limited in two aspects. First, the heuristically constructed paper association graphs used for representation learning contain uncertainties that may cause negative supervision. Second, existing algorithms, such as binary cross-entropy loss, used to train representation learning models may not produce sufficiently high-quality representations for AND. To tackle the above problems, we propose an association refining and compositional contrasting (ARCC) framework for AND tasks. ARCC first adopts an iterative graph structure refinement process to dynamically reduce the uncertainties in paper graphs. Then, a compositional contrastive learning method is proposed to encourage learning more discriminative representations for AND. Empirical studies on two benchmark datasets suggest that ARCC is effective for AND and outperforms the state-of-the-art models.

## CCS CONCEPTS

• **Information systems** → **Information retrieval**; *Document representation.*

## KEYWORDS

Author Name Disambiguation, Graph Structure Refinement, Contrastive Learning

### ACM Reference Format:

Anonymous Author(s). 2024. Author Name Disambiguation via Paper Association Refinement and Compositional Contrastive Embedding. In *Proceedings of the Web (Conference WWW 2024)*. ACM, New York, NY, USA, 10 pages. https://doi.org/XXXXXXX.XXXXXXX

## 1 INTRODUCTION

The author name disambiguation (AND) problem, i.e. authors sharing the same name, is one of the crucial phenomenons in online academic retrieval systems such as DBLP [3] and GoogleScholar [6]. The presence of disambiguation introduces noise during the assignment of papers to their respective authors, thus posing challenges in precisely accessing web content.

To alleviate the AND phenomenon, most of the existing models develop representation methods to characterize the underlying

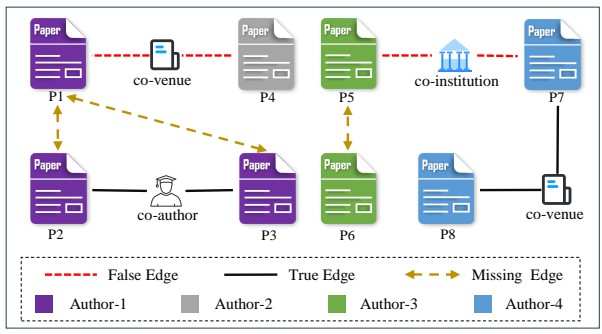

**Figure 1: Examples illustrating the uncertainty in paper association graph. (1) False Edge: the error edges caused by misleading associations. (2) Missing Edge: The edges are missing because no explicit associations detected.**

features of papers and then translate this task into a paper similarity modelling problem. One of the resources that can be used to deal with AND problem is the textual features from the title, abstract, or other content in the papers. Textural features are usually extracted by existing text encoders, such as Word2vec [18] and Doc2vec [15]. In addition to the textural feature, the associations, such as co-author, co-institution, and co-venue, between papers, are further utilized to learn better paper representations. These associations are leveraged to construct a paper association graph, where *papers of the same author are expected to be connected by edges*. Based on this graph, structural features are extracted through commonly used graph encoders, such as Node2vec [8], GCN [14], and GAT [22]. The inclusion of structural information offers additional resources to enhance the performance of the AND model.

However, since the paper association graph is built heuristically [23, 26, 29–32], it may inevitably contain uncertainties. These uncertainties may introduce two types of error edges, i.e., false edge and missing edge. For example, in Figure 1, the heuristic method establishes a connection between paper P1 and P4, as they share a common venue. Similarly, papers P5 and P7 are linked together due to they share the same institution. However, the two connections are unreal because the two linked papers are authored by different individuals. We call such connections misleading associations, which cause false edges among paper nodes. Figure 1 also shows the missing edges between paper P1 and P2, P1 and P3, and P5 and P6. These missing edges are caused by the absence of explicit associations, such as co-author, co-venue, etc. The presence of error edges stemming from uncertainties introduces noise in paper representation learning, thereby degrading the representativeness of the paper embedding. Consequently, the degradation adversely affects the performance of author name disambiguation.

The above analysis inspires us that narrowing down the gap between the constructed uncertain association graph and the ground truth association graph is one of the key factors that need to be considered for enhancing author name disambiguation. To achieve a more reliable paper association graph, we propose a paper association refinement approach to eliminate uncertainties. Specifically, we present a two-stage learning strategy, which first heuristically refines the paper association graphs by defined judgmental conditions and then dynamically updates the structure of the paper association graphs iteratively based on semantic embedding. The two-stage refinement helps reduce false edges and recover missing edges in the paper association graph, thus providing more reliable structure information for the AND task.

Furthermore, high-quality representations play a crucial role in enhancing similarity measurement and improving the performance of paper clustering in the AND task. To improve the representation quality, we propose a novel approach called compositional contrastive embedding, which aims to enhance the semantic embedding, structure embedding, and fused representation. Concretely, we employ a sampling strategy where papers authored by the same individual are selected as positive or negative pairs based on their associated labels. Subsequently, we utilize the semantic embedding, structure embedding, and fused embedding of the sampled pairs for supervised contrastive learning. This compositional contrastive embedding method significantly improves the discriminativeness of both the semantic-level and structure-level representations, as well as the fused representations. As a result, these representations become more effective in generalizing to unknown test papers in the AND task. Additionally, we leverage the learned semantic embedding to iteratively refine the association graph, which further enhances the quality of the association graph, leading to improved performance overall.

The whole framework, which combines the paper association refinement approach with the compositional contrastive embedding approach, is named as ARCC. We conduct empirical studies on the most commonly used AND datasets, i.e., AMiner-18 and WhoisWho-SND, and the results confirm the effectiveness of our proposed iterative AND model[1].

To summarize, our main contributions are in the following three folds:

- We propose paper association refinement approach to eliminate uncertainties paper association, which helps reduce false edges and recover missing edges in the paper association graph.
- We present a compositional contrastive learning module to fuse the textual and structural paper features and adaptively learn the importance of these features. The module further introduced two separate contrastive losses for both semantical-level and structural-level features to increase the generalization of our iterative model.
- Empirical studies indicate that our proposed model outperforms previous state-of-the-art methods on two real-world datasets, and comprehensive analysis confirms the effectiveness of our framework.

---

[1]Code is released at https://anonymous.4open.science/r/ARCC-66CB

## 2 RELATED WORK

Author name disambiguation (AND) is a challenging task that aims to distinguish authors of the same name, which is crucial in improving the user experience of online academic retrieval systems. Existing AND methods mainly involve learning paper representations computing their similarity, and finally clustering them. The solutions for AND task can be divided into two categories: pairwise classification methods, and graph-based methods.

The pairwise classification methods translate the AND problem into a pairwise similarity learning problem. The similarity among papers is computed according to their features which are based on their textual [9, 10, 16, 27, 31]. These approaches are based on discriminative models and pairwise classifiers are trained to disambiguate author names. Louppe et al. [16] exploited a supervised classifier based on handcrafted features to learn pairwise similarities and semi-supervised hierarchical clustering models are utilized to generate final clusters. Zhang et al. [31] studied the disambiguation results with and without incorporating neural networks where two sets of heuristic rules and simple neural networks are incorporated to merge papers. However, pairwise classification methods do not fully utilize the features between papers and may not directly capture the relation between the papers, resulting in insufficient generalization of the model.

Graph-based methods are currently the most favored approach. The main idea is to construct an association graph of the papers and jointly learn the structure and semantic information [5, 17, 19, 20, 23, 28–30, 32]. Zhang and Al Hasan [28] designed three types of association graphs and concatenated these features into one single paper embedding. Zhang et al. [29] uses a graph convolutional network (GCN) to incorporate the global semantical and the local structural features of the paper association graph. Wang et al. [23] borrowed the adversarial learning model, where the discriminator determines if two papers are from the same author and the generator chooses similar papers from a heterogeneous network. Zhang et al. [30] jointly learned the semantical and structural features by optimizing the paper embedding via a GCN model. Zhou et al. [32] built multiple-type graphs and introduced graph learning to obtain multiple feature information. These features are fed into a GCN-like model for feature integration. Pooja et al. [19] applied an attention-based multi-dimensional multi-hop neighborhood-based graph convolution network to represent documents in heterogeneous graphs. Although these Graph-based methods improve AND by incorporating graph structural information, they ignore the uncertainty in the paper association graphs.

Traditional contrastive learning is an unsupervised method that learns representations by pulling close of positive pairs and pushing away those of negative pairs [2, 24]. Inspired by previous studies, Khosla et al. [11] proposed a new method called SupCon for supervised contrastive learning. This method aims to bring normalized embeddings from the same class closer together while pushing embeddings from different classes further apart. It utilizes label information to achieve this. Supcon achieved better results than standard CE minimization, which has been shown to be more generalizable and robust to noisee [7]. In this paper, we also introduce contrastive learning to enhance paper representations based on the semantic and structure infromation.

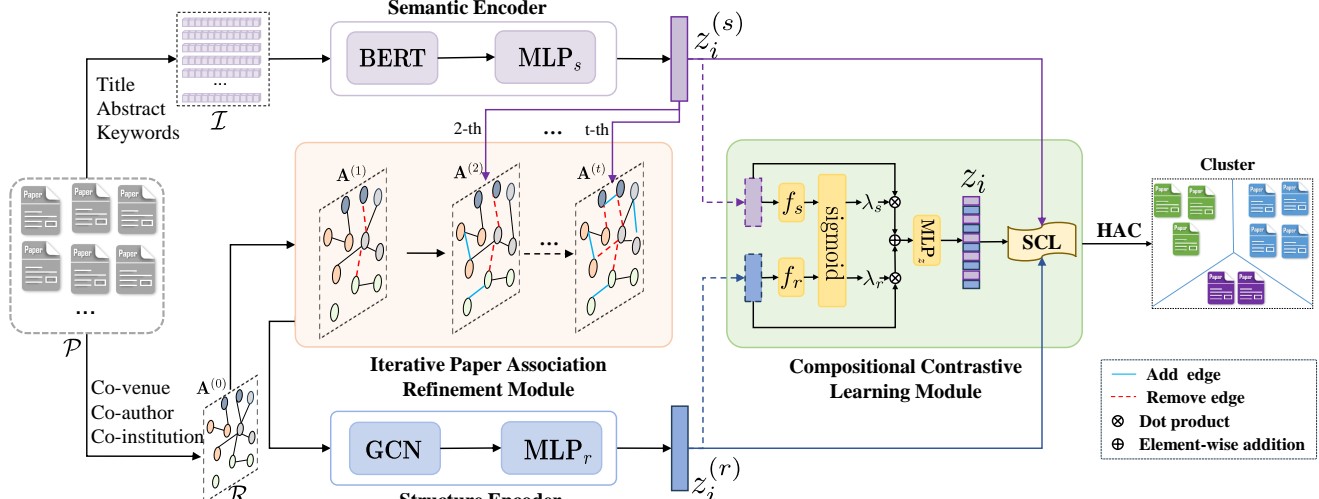

**Figure 2: The architecture of the association refining and compositional contrasting framework. The encoder is used to obtain the semantic embedding $z_i^{(s)}$ and structure embedding $z_i^{(r)}$. The paper association refinement module iteratively updates the graph structure $A^{(t)}$. The paper embedding $z_i$ is computed by the compositional contrastive learning module and fed to HAC for clustering.**

## 3 METHODOLOGY

### 3.1 Problem Formulation

Given an author name reference $a$ and a group-size $K$, assume that we have $N$ papers $\mathcal{P}^a = \{P_1^a, P_2^a, ..., P_N^a\}$ belonging to authors with the same name $a$. Author name disambiguation aims to divide the $N$ papers into $K$ groups, each corresponding to an individual author. For each paper $P_i^a \in \mathcal{P}^a$, a textual attribute set $\mathcal{I}_i^a$ is given, which includes title, abstract, and keywords. Additionally, we may construct a potential structural feature set $\mathcal{R}_i^a$ that indicates co-author, co-institution, and co-venue relations of authors. With these resources, an AND model tries to find a function $\Phi$ to cluster $\mathcal{P}^a$ into disjoint collections $\{C_i^a\}_{i=1}^K$ given the associated paper set $\mathcal{P}^a$, the text attribute set $\mathcal{I}_i^a$ and the constructed $\mathcal{R}_i^a$ as follows.

$$\Phi(\mathcal{P}^a | \mathcal{I}^a, \mathcal{R}^a) \rightarrow \{C_1^a, C_2^a, ..., C_K^a\} \tag{1}$$

where $C_k^a \subset \mathcal{P}^a$ is expected to contain all papers in paper set $\mathcal{P}^a$ that belongs to the $k^{\text{th}}$ author with name $a$. *We will omit the superscript $a$ for simplicity in later use.*

### 3.2 Overall Framework

We propose an association refining and compositional contrasting (ARCC) framework illustrated in Figure 2, which consists of three components: feature encoder, iterative paper association refinement module, and compositional contrastive learning module. The feature encoder consists of a semantic encoder and a structure encoder respectively for semantic and structure embeddings. The iterative paper association refinement module is designed to achieve a more reliable paper association graph. The compositional contrastive learning module generates better paper representations by applying feature-level contrasting and compositional contrasting.

### 3.3 Feature Encoder

*3.3.1 Semantic Encoder.* To obtain rich semantic representations from the textual feature, we utilize the power of pre-trained models by employing BERT [4] as the semantic encoder. Specifically, we introduce a function $\textbf{BERT}(\cdot, \cdot)$ to denote the BERT encoding process, which treats the embedding corresponding to [CLS] as the output. Let function $\textbf{MLP}_s(\cdot, \cdot)$ denote a 2-layer MLP with Relu$(\cdot)$ activation, then the semantic encoder can be formulated as:

$$\mathbf{Z}^{(s)} = \textbf{MLP}_s(\textbf{BERT}(\mathcal{I}; \theta_s); \phi_s) \tag{2}$$

where $\mathbf{Z}^{(s)} = \{z_1^{(s)}, z_2^{(s)}, \cdots, z_n^{(s)}\}$ denotes semantic embedding of all papers in $\mathcal{P}$. $\phi_s$ and $\theta_s$ denote parameters of the MLP and BERT.

*3.3.2 Structure Encoder.* Another essential information important for AND is the potential structural features $\mathcal{R}$ indicating collaborative relationships among papers, work backgrounds, and academic communications. These structural features are from a *paper association graph*. The initial graphs are heuristically constructed based on the knowledge of the co-occurrence of authors, institutes, and venues. Specifically, we define the initial paper association graph as $\mathcal{G}(\mathbf{X}, \mathbf{A})$, where $\mathbf{A} \in \{0, 1\}^{N \times N}$ is the adjacent matrix and $\mathbf{A}_{i,j} = 1$ indicates co-occurrence relations including co-author, co-institution, and co-venue between $P_i$ and $P_j$ [26]. $\mathbf{X} \in \mathcal{R}^{N \times d}$ is the node representations initialized by the Node2Vec [8].

To obtain a better representation of the paper association graphs, we utilize GCN [13] as a structure encoder to capture high-order structural features. The final structure embedding is generated by the GCN followed by a 2-layer MLP, which is formulated as:

$$\mathbf{Z}^{(r)} = \textbf{MLP}_r(\textbf{GCN}(\mathbf{X}, \mathbf{A}; \theta_r); \phi_r) \tag{3}$$

where $\mathbf{Z}^{(r)} = \{z_1^{(r)}, z_2^{(r)}, \cdots, z_n^{(r)}\}$ denotes the structure embedding. $\phi_r$ and $\theta_r$ are the parameters of the MLP and GCN.

## 3.4 Iterative Paper Association Refinement Module

As discussed above, the initial paper association graph is constructed heuristically. Although this allows us to easily obtain an initial graph from scratch, it also brings uncertainties into the graph structure. Thus, we need to refine the graph structure during training to learn better structure representations. We propose a rule-guided graph and semantic-guided graph refinement module to update the structure of the paper association graph.

*3.4.1 Rule-Guided Graph Refinement.* Let $\mathbf{A}^{(0)}$ denote the initial paper association graph. We first heuristically reduce the potential false edges in the initial graph. We define this rule-guided graph refinement process as follows:

$$\mathbf{A}^{(1)} = \mathbb{I}(\mathcal{H}) \wedge \mathbf{A}^{(0)} \tag{4}$$

where $\wedge$ represents the logic operation AND and $\mathbb{I}(\cdot)$ is an indicator function. $\mathcal{H}$ represents the heuristic judgmental conditions, including the frequency of collaboration of co-authors, the intuition of a research team from the same institution, and the similarity of the authors' research topics (see Appendix for details).

*3.4.2 Semantic-Guided Graph Refinement.* Removing potential false edges using a rule-guided approach may be not sufficient to build a reliable-enough graph structure because the missing edges are not considered. During training, the structure embedding is jointly optimized with semantic embedding, and the node embeddings are correspondingly updated. This allows us to dynamically refine the graph structure with the assistance of semantic embedding.

In more detail, we first obtain semantic embedding $\mathbf{Z}^{(s)}$ through the Semantic Encoder. Then, we design an edge-dropping function $\mathcal{T}_{de}(\cdot, \cdot, \cdot)$ to drop the low semantic similarity edges between $P_i$ and $P_j$ from the graph. Correspondingly, we introduce an edge-adding function $\mathcal{T}_{ae}(\cdot, \cdot, \cdot)$ to dynamically add high-confidence edges. The dynamic refinement process can be formulated as follows:

$$\mathbf{A}_{i,j}^{(t+1)} = \mathbf{A}_{i,j}^{(t)} \vee \mathcal{T}_{ae}(z_i^{(s)}, z_j^{(s)}, \gamma) \wedge \neg \mathcal{T}_{de}(z_i^{(s)}, z_j^{(s)}, \psi) \tag{5}$$

where $\mathbf{A}^{(t+1)}$ is more reliable graph structure, $t$ starts from 1. $\neg$ represents the logic NOT , and $\vee$ represents the logic OR. The function $\mathcal{T}_{ae}$ and $\mathcal{T}_{de}$ are specified as follows:

$$\mathcal{T}_{ae}(z_i^{(s)}, z_j^{(s)}, \gamma) = \begin{cases} 1, & \text{if} \quad s(z_i^{(s)}, z_j^{(s)}) \geq \gamma \\ 0, & \text{otherwise.} \end{cases} \tag{6}$$

$$\mathcal{T}_{de}(z_i^{(s)}, z_j^{(s)}, \psi) = \begin{cases} 1, & \text{if} \quad s(z_i^{(s)}, z_j^{(s)}) \leq \psi \\ 0, & \text{otherwise.} \end{cases} \tag{7}$$

where $z_i^{(s)}$ and $z_j^{(s)}$ is semantic embedding of $P_i$ and $P_j$. $\gamma$ and $\psi$ are hyperparameters that indicate the lower threshold to add an edge and the upper threshold to drop an edge, respectively. Specifically, if $s(z_i^{(s)}, z_j^{(s)}) \geq \gamma$, the model will add a new edge between $P_i$ and $P_j$, where $s(\cdot, \cdot)$ is a cosine similarity function in Eq. (9). Conversely, if $s(z_i^{(s)}, z_j^{(s)}) \leq \psi$, the model will remove an edge.

In this way, the graph structure can be dynamically updated with the semantic embedding changes during training, which enables the model to obtain more reliable graph structures by reducing false edges and recovering missing edges.

## 3.5 Compositional Contrastive Learning Module

To obtain better semantic and structure embeddings and improve their generalization abilities to unknown user names in the test set, we propose a compositional contrastive learning module, which respectively performs feature-level contrastive learning with semantic and structure embedding and when they are composed as the unified embedding.

*3.5.1 Feature-level Contrastive Embedding.* Inspired by contrastive learning to obtain better representations [11], we adopt a feature-level contrasting representation approach to train the model to increase the generalization of ARCC, Specifically, given semantic embedding $\mathbf{Z}^{(s)}$ and structure embedding $\mathbf{Z}^{(r)}$, we optimize the representation of each feature by feature-level contrastive loss, which is formulated as follows:

$$\mathcal{L}_f = -\frac{1}{n} \sum_i^n \frac{1}{|G|} \sum_{j \in G} (\log \frac{\exp(s(z_i^{(s)}, z_j^{(s)})/\tau_s)}{\sum_A \exp(s(z_i^{(s)}, z_a^{(s)})/\tau_s)} + \alpha \log \frac{\exp(s(z_i^{(r)}, z_j^{(r)})/\tau_r)}{\sum_A \exp(s(z_i^{(r)}, z_a^{(r)})/\tau_r)}) \tag{8}$$

where $n$ is the number of papers with the same name, and $\alpha$ is a weight factor. $G$ is the group of papers written by the same author, $|G|$ represents the number of papers by that author, and $A$ is a collection of papers with the same name. $\tau_s$ and $\tau_r$ are hyperparameters for temperature factors of different features to mining hard negative instances. The similarity between the two instances is measured using cosine distance denoted as $s(\cdot, \cdot)$, following the method used in NT-Xent [1]. The $\langle \cdot, \cdot \rangle$ donated the dot product operator.

$$s(x_i, x_j) = \frac{\langle x_i, x_j \rangle}{\|x_i\| \|x_j\|} \tag{9}$$

*3.5.2 Compositional Contrastive Embedding.* We introduced the compositional contrastive representation approach to fuse the semantic features and structure features and adaptively learn the importance of these features.

First, we obtained the semantic embedding $\mathbf{Z}^{(s)}$ and structure embedding $\mathbf{Z}^{(r)}$ by the Eq. (2) and Eq. (3) respectively. Then, we introduce two weights $\lambda_s$ and $\lambda_r$ to the adaptive composition of the discriminative feature of the two features, which can be computed by:

$$\lambda_s = \text{sigmoid}(f_s(\mathbf{Z}^{(s)}, W_s)) \tag{10}$$

$$\lambda_r = \text{sigmoid}(f_r(\mathbf{Z}^{(r)}, W_r)) \tag{11}$$

$$\lambda_s + \lambda_r = 1 \tag{12}$$

where $f_s(\cdot, \cdot)$ and $f_r(\cdot, \cdot)$ are the MLP function, and $W_s$ and $W_r$ are the parameters to be trained. The Eq. (12) is the constraint on the weight values. Finally, we can obtain compositional paper embedding based on weights as follows:

$$\mathbf{Z} = \text{MLP}_z(\lambda_s \mathbf{Z}^{(s)} + \lambda_r \mathbf{Z}^{(r)}, \phi_z) \tag{13}$$

where $\mathbf{MLP}_z(\cdot, \cdot)$ is the projection function using 2-layer MLP, and $\phi_z$ is the parameter that can be trained by compositional contrastive loss. The loss function can be formulated as:

$$\mathcal{L}_z = -\frac{1}{n} \sum_i^n \frac{1}{|G|} \sum_{j \in G} \log \frac{\exp(s(z_i, z_j)/\tau_z)}{\sum_A \exp(s(z_i, z_a)/\tau_z)} \qquad (14)$$

where $\tau_z$ is the temperature factor. The overall objective $\mathcal{L}$ is combined with two losses, including the feature-level contrastive losses, and the compositional contrastive loss:

$$\mathcal{L} = \mathcal{L}_f + \beta \mathcal{L}_z \qquad (15)$$

where $\beta$ are hyper-parameters that balance the weight of the different losses. The compositional paper embedding $\mathbf{Z} \in \mathbb{R}^{N \times d}$ captures the disambiguation-relevant features from semantic and structure embedding.

## 3.6  Clustering

After obtaining compositional paper embedding, our author name disambiguation framework clusters $N$ papers into $K$ groups. With paper similarity function $s(\cdot, \cdot)$ defined as Eq. (9), we equivalently define the distance between two papers as $d(P_i, P_j) = 1 - s(P_i, P_j)$. Besides this sample-sample distance, clustering algorithms typically need to measure cluster-cluster distance or sample-cluster distance to reflect inter-cluster difference. These distances can be computed based on sample-sample distance. For example, to calculate the distance $d(C_k, C_{k+1})$ between two clusters $C_k, C_{k+1}$, we adopt the *average linkage criterion*. That is, $d(C_k, C_{k+1})$ equals the average over distances $d(P_i, P_j)$ of all pairs $\{(P_i, P_j) | P_i \in C_k, P_j \in C_{k+1}\}$.

To fairly compare with previous works [23, 28], we exploit the hierarchy agglomerative clustering (HAC) algorithm for paper clustering [25]. Specifically, we calculate all the distances between pairs of $N$ papers with the same ambiguous name to form a distance matrix $\mathbf{D} \in \mathbb{R}^{N \times N}$, where $\mathbf{D}_{ij} = d(P_i, P_j)$. Given the number $K$ of clusters, HAC takes the precomputed distance matrix $\mathbf{D}$ as input, recursively merges pairs of clusters of papers, and finally returns the result of the clustering assignment for each paper.

## 4  EXPERIMENT

### 4.1  Experimental Setup

*4.1.1  Datasets.* Two commonly-used real-world author name disambiguation datasets, AMiner-18 [2] and WhoisWho-SND [3], are used for evaluating the performance of our proposed ARCC framework. Table 1 shows the detailed statistics of these two datasets.

**AMiner-18** contains 600 author names including 400 author names for training 100 author names for validating, and 100 names for testing. Title, abstract, keywords, and publication information are associated with each paper. Our training and testing splitting follows the one presented in [29].

**WhoisWho-SND** contains 220, 51, and 50 author name references for training, validating, and testing respectively. The title, abstract, keywords, publication venue and year, author names, and institutions are also associated with each paper.

[2]https://static.aminer.cn/misc/na-data-kdd18.zip
[3]https://www.aminer.cn/billboard/whoiswho

**Table 1: The statistics of AMiner-18 and Who-sWho datasets.**

|  | Aminier-18 | WhoisWho-SND |
|---|---|---|
| # of names | 600 | 321 |
| # of authors | 39781 | 26093 |
| # of papers | 208827 | 292448 |
| # of authors per name min/max/avg | 2/542/66.3 | 0/588/81.3 |
| # of papers per name min/max/avg | 192/916/348.0 | 0/5682/911.1 |

*4.1.2  Evaluation Metrics.* To evaluate the results of this unsupervised clustering task, the existing technique is to evaluate a series of binary classification tasks. An author has written papers $C_i \subset \mathcal{P}$, if and only if each pair of papers in $C_i$ are written by the same author, and each pair of papers one from $C_i$ and the other from its complement set $\mathcal{P} - C_i$ are not written by a same author. By determining whether each of the $\binom{N}{2}$ pairs of papers are written by one author, we equivalently solve the author name disambiguation clustering task. Thus we can then evaluate the task using pairwise Precision (Pre), pairwise Recall (Rec), and pairwise F1-score (F1) [23, 29] (see Appendix for details).

*4.1.3  Implementation Details.* We use PyTorch for implementation. For all experiments, the hidden dimension is chosen as 100, the number of training epochs is 150, and the dropout ratio is set as 0.5. Batch size is determined by the number of papers belonging to the same name reference.

For model training, we use grid search to obtain the best hyper-parameters, which include the learning rate selected from {0.001, 0.005, 0.0001, 0.0005}, the contrastive learning temperature $\tau$ selected from the range of 0.025 to 0.50, and the loss weight factor $\alpha$ and $\beta$ selected from {0.25, 0.5, 0.75, 1.0}, and the lower threshold for adding an edge $\gamma$ selected from 0.50 to 0.975, and the upper threshold for dropping an edge $\psi$ selected from 0.05 to 0.475. The Adam optimizer [12] is exploited to optimize model parameters, and all calculations are done on four NVIDIA Tesla V100 GPUs.

*4.1.4  Baselines.* In order to verify the effectiveness of our proposed ARCC framework, we conducted a comparison study between our model and other existing author name disambiguation methods, including Beard [16], AGAND [28], AMiner [29], GANAND [23], MFAND [32], MGATAND [30], and MRAND [19]. Note that all reported results for AMiner-18 dataset are taken from the corresponding original paper. For WhoisWho-SND dataset, we download the source code and report the results of our re-implementation.

## 4.2  Overall Results

Table 2 shows the results of our proposed ARCC and other comparing baselines on two datasets. We compute the macro average score, and the result of our model is an average of 5 runs of experiments.

It can be observed that ARCC outperforms all other baseline models with a great margin (at least 4% and 3% in F1-score) on both Amier-18 and WhoisWho-SND. These results confirm the effectiveness of our ARCC model for the AND task.

Another observation is that our model achieves a lower Pre than some existing models and significantly improves the Rec on

**Table 2: The performance comparison of different baselines**

| AMiner-18 | | | |
|---|---|---|---|
| **Model** | **Pre** | **Rec** | **F1** |
| Beard [16] | 57.09 | 77.22 | 63.10 |
| AGAND [28] | 70.63 | 59.53 | 62.81 |
| AMiner [29] | 77.96 | 63.03 | 67.79 |
| GANAND [23] | 82.23 | 67.23 | 72.92 |
| MGATAND [30] | **83.87** | 64.91 | 73.10 |
| MFAND [32] | 81.39 | 69.47 | 74.92 |
| MRAND [19] | 72.40 | 75.10 | 71.50 |
| ARCC | $78.09_{\pm1.25}$ | $\mathbf{82.32_{\pm0.70}}$ | $\mathbf{78.96_{\pm0.53}}$ |
| **WhoisWho-SND** | | | |
| **Model** | **Pre** | **Rec** | **F1** |
| Beard [16] | 72.20 | 46.19 | 56.34 |
| AGAND [28] | 76.40 | 35.20 | 48.19 |
| AMiner [29] | 77.70 | 55.50 | 64.75 |
| MGATAND [30] | 67.98 | 79.99 | 73.45 |
| MFAND [32] | 73.36 | 81.03 | 77.00 |
| ARCC | $\mathbf{79.52_{\pm0.74}}$ | $\mathbf{83.15_{\pm0.48}}$ | $\mathbf{80.11_{\pm0.57}}$ |

AMiner-18. We consider the Rec as a more critical indicator for evaluating the disambiguation capability of algorithms, especially in the AND task, a higher Rec metric indicates that more papers from one single author form one single cluster, representing the global disambiguation performance.

For a detailed look at these results, Table 4 shows the results of 15 name references sampling from AMiner-18. Our model improves almost all the evaluation metrics. These results confirm the stabilized performance of our proposed ARCC model.

## 4.3 The Effectiveness of the Iterative Refinement Process

*4.3.1 Ablation Analysis.* To evaluate the effectiveness of the components of the iterative paper association refinement module, we created the following three variants for comparison.

$\text{ARCC}_{\text{w/o IPR}}$ removes the iterative paper association refinement module (IPR) and the initial paper association graph structure is not updated.

$\text{ARCC}_{\text{w/o SR}}$ removes the semantic-guided graph refinement component (SR) and keeps our rule-guided graph refinement model to capture paper features.

$\text{ARCC}_{\text{w/o RR}}$ removes the rule-guided graph refinement component (RR) and keeps the semantic-guided graph refinement module to characterize the paper features.

The ablation analysis results are shown in Table 3, It can be observed that removing IPR decreases the performance of ARCC by 3.6% and 3.3% on both datasets. These facts reveal that the initial graph structure may contain unreliable or missing associations. We believe that our presented dynamic model generates more reliable structural representations. Secondly, $\text{ARCC}_{\text{w/o SR}}$ decrease the performance of ARCC by 1.7% and 1.3%. These results indicate that

**Table 3: Ablation study on iterative paper association refinement module on AMiner-18 and WhoisWho-SND**

| AMiner-18 | | | |
|---|---|---|---|
| **Model** | **Pre** | **Rec** | **F1** |
| $\text{ARCC}_{\text{w/o IPR}}$ | $74.17_{\pm0.83}$ | $79.70_{\pm0.68}$ | $75.33_{\pm0.39}$ |
| $\text{ARCC}_{\text{w/o RR}}$ | $75.17_{\pm0.46}$ | $80.11_{\pm0.34}$ | $75.98_{\pm0.22}$ |
| $\text{ARCC}_{\text{w/o SR}}$ | $75.38_{\pm0.32}$ | $81.62_{\pm0.59}$ | $76.95_{\pm0.23}$ |
| ARCC | $\mathbf{78.09_{\pm1.25}}$ | $\mathbf{82.32_{\pm0.70}}$ | $\mathbf{78.96_{\pm0.53}}$ |
| **WhoisWho-SND** | | | |
| $\text{ARCC}_{\text{w/o IPR}}$ | $77.89_{\pm0.86}$ | $78.78_{\pm1.16}$ | $76.80_{\pm0.45}$ |
| $\text{ARCC}_{\text{w/o RR}}$ | $77.37_{\pm0.69}$ | $80.38_{\pm1.04}$ | $77.40_{\pm0.44}$ |
| $\text{ARCC}_{\text{w/o SR}}$ | $\mathbf{80.56_{\pm0.75}}$ | $78.15_{\pm1.02}$ | $77.94_{\pm0.55}$ |
| ARCC | $79.52_{\pm0.74}$ | $\mathbf{83.15_{\pm0.48}}$ | $\mathbf{80.11_{\pm0.57}}$ |

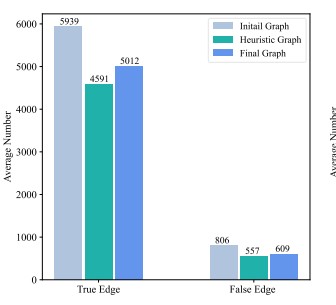

(a) Comparison of refining stages on test dataset

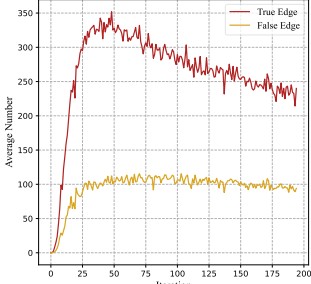

(b) Iterative refining process on training dataset

**Figure 3: The statistics on the quantity of data for structure refining on AMiner-18**

the rule-guided graph refinement module increases the reliability of the original paper associations.

It is also worth noting that both $\text{ARCC}_{\text{w/o RR}}$ and $\text{ARCC}_{\text{w/o SR}}$ outperform $\text{ARCC}_{\text{w/o IPR}}$ on both datasets. This validates the effectiveness of our proposed framework.

*4.3.2 Quantitative Analysis of Structure Refinement.* To validate the reliability of the updated associations between papers, we compare the paper association graph structure with the ground truth.

Figure 3 (a) shows the overlapping statistics of the training process on AMiner-18. It can be observed that the initial graph has many false edges, leading to noises for further representation learning. The rule-guided graph has fewer false edges and also contains fewer true edges, indicating that the rule-guided refinement removes true edges while removing many false edges. Finally, the final graph combines the initial graph and rule-guided graph to increase true edges while minimizing false edges. The experimental results show that our proposed ARCC effectively reduces false edges while mitigating noise impact, leading to improved performance despite reducing some true edges.

Figure 3(b) shows a detailed association graph coverage during the iterative training process. It can be observed a steep curve at the

**Table 4: The detailed results on AMiner-18**

| Name | ARCC | | | MFAND [32] | | | AMiner [29] | | | Beard [16] | | | AGAND [28] | | |
|---|---|---|---|---|---|---|---|---|---|---|---|---|---|---|---|
| | Pre | Rec | F1 | Pre | Rec | F1 | Pre | Rec | F1 | Pre | Rec | F1 | Pre | Rec | F1 |
| xu_xu | 65.15 | 69.25 | **67.14** | 34.59 | 79.44 | 48.20 | 74.18 | 45.86 | 56.68 | 48.16 | 41.87 | 44.80 | 22.55 | 64.40 | 33.40 |
| rong_yu | 97.58 | 97.96 | **97.77** | 72.31 | 43.83 | 54.58 | 89.13 | 46.51 | 61.12 | 65.48 | 40.85 | 50.32 | 38.85 | 91.43 | 54.53 |
| yong_tian | 64.85 | 76.68 | **70.27** | 46.12 | 59.42 | 51.93 | 76.32 | 51.95 | 61.82 | 70.74 | 56.85 | 63.04 | 32.08 | 63.71 | 42.67 |
| lu_han | 53.25 | 46.28 | **49.52** | 37.22 | 51.25 | 43.12 | 51.78 | 28.05 | 36.39 | 47.88 | 20.62 | 28.82 | 30.25 | 46.65 | 36.70 |
| lin_huang | 82.98 | 59.02 | **68.98** | 60.80 | 52.40 | 56.29 | 77.10 | 32.87 | 46.09 | 71.84 | 34.17 | 46.31 | 24.86 | 71.32 | 36.87 |
| kexin_xu | 91.41 | 98.51 | 94.83 | 83.50 | 81.93 | 82.71 | 91.37 | 98.64 | **94.87** | 90.02 | 82.47 | 86.08 | 91.26 | 98.35 | 94.67 |
| wei_quan | 91.03 | 92.61 | **91.81** | 35.72 | 48.67 | 41.20 | 53.88 | 39.02 | 45.26 | 64.45 | 47.66 | 54.77 | 37.86 | 63.41 | 47.41 |
| tao_deng | 79.54 | 68.23 | **73.45** | 59.55 | 41.22 | 48.72 | 81.63 | 43.62 | 56.86 | 53.04 | 29.89 | 38.23 | 40.46 | 51.38 | 45.27 |
| hongbin_li | 76.87 | 93.89 | **84.53** | 48.85 | 78.86 | 60.33 | 77.20 | 69.21 | 72.99 | 54.66 | 53.05 | 53.84 | 19.48 | 85.96 | 31.77 |
| hua_bai | 83.52 | 90.86 | **87.04** | 73.66 | 55.82 | 63.51 | 71.49 | 39.73 | 51.08 | 58.58 | 35.90 | 44.52 | 36.39 | 41.33 | 38.70 |
| mei_ling_chen | 58.47 | 85.70 | **69.52** | 94.80 | 41.78 | 58.00 | 74.93 | 44.70 | 55.99 | 59.36 | 28.80 | 38.79 | 58.32 | 47.14 | 52.14 |
| yanqing_wang | 31.11 | 88.96 | 46.10 | 70.37 | 57.20 | 63.10 | 71.52 | 75.33 | **73.37** | 60.40 | 51.97 | 55.87 | 29.64 | 79.08 | 43.11 |
| xu_dong_zhang | 86.13 | 59.89 | 70.66 | 51.48 | 24.17 | 32.90 | 62.40 | 22.54 | 33.12 | 70.20 | 23.35 | 35.04 | 72.38 | 79.83 | **75.92** |
| qiang_shi | 54.46 | 54.42 | 52.90 | 40.53 | 76.46 | **52.97** | 52.20 | 36.15 | 42.72 | 43.84 | 36.94 | 40.10 | 35.31 | 47.18 | 40.39 |
| min_zheng | 77.05 | 49.95 | **60.61** | 31.74 | 52.48 | 39.55 | 57.65 | 22.35 | 32.21 | 54.76 | 19.70 | 28.98 | 25.86 | 32.67 | 28.87 |

first few initial iterations. This suggests that high-quality semantic representations will benefit the similarity measurement.

The above observations support our motivation to perform better via a more reliable graph structure so that the gap between the constructed association and the ground truth association are then narrowed down.

### 4.4 Effectiveness of Compositional Contrastive Learning Module

*4.4.1 Ablation Analysis.* To explore the effectiveness of the compositional contrasting learning module, we designed three model variants.

**ARCC**$_{w/o\ CCL}$ removes the compositional contrastive learning module (CCL) and optimizes the model using binary cross-entropy loss.

**ARCC**$_{w/o\ FC}$ removes the feature-level contrastive embedding component (FC), which is optimized using only the compositional contrastive loss.

**ARCC**$_{w/o\ CC}$ removes the compositional contrastive embedding component (CC), which directly concatenates semantic and structure embedding as paper embedding.

It can be observed that the performance of the ARCC $_{w/o\ CCL}$ compared to the ARCC decreases by at least 5% for both datasets in Table 5. The result shows that contrast learning can force the model to learn a more discriminative representation than the pairwise supervised method. Secondly, the performance of ARCC$_{w/o\ FC}$ compared to ARCC decreases by at least 4.6% in both datasets, which illustrates the effectiveness of feature-level contrastive representation. Then, the performance of ARCC$_{w/o\ CC}$ decreases by 4.3% and 1.5% on AMiner-18 and WhoisWho-SND, respectively, which indicates that different features of each disambiguation instance have different importance, and the adaptively compositional approach obtain better representation. Finally, these results suggest that the feature-level contrastive embedding and compositional contrastive

**Table 5: Ablation study on compositional embedding**

| AMiner-18 | | | |
|---|---|---|---|
| Model | Pre | Rec | F1 |
| ARCC$_{w/o\ CCL}$ | $71.46_{\pm1.55}$ | $80.08_{\pm0.69}$ | $73.79_{\pm0.87}$ |
| ARCC$_{w/o\ FC}$ | $70.66_{\pm0.91}$ | $82.21_{\pm0.50}$ | $74.32_{\pm0.76}$ |
| ARCC$_{w/o\ CC}$ | $74.52_{\pm0.59}$ | $78.10_{\pm0.52}$ | $74.65_{\pm0.58}$ |
| ARCC | $\mathbf{78.09_{\pm1.25}}$ | $\mathbf{82.32_{\pm0.70}}$ | $\mathbf{78.96_{\pm0.53}}$ |
| **WhoisWho-SND** | | | |
| ARCC$_{w/o\ CCL}$ | $68.51_{\pm0.65}$ | $76.21_{\pm0.75}$ | $69.79_{\pm0.67}$ |
| ARCC$_{w/o\ FC}$ | $70.84_{\pm1.55}$ | $78.09_{\pm0.95}$ | $72.54_{\pm1.24}$ |
| ARCC$_{w/o\ CC}$ | $\mathbf{80.97_{\pm1.15}}$ | $79.11_{\pm1.61}$ | $78.55_{\pm0.90}$ |
| ARCC | $79.52_{\pm0.74}$ | $\mathbf{83.15_{\pm0.48}}$ | $\mathbf{80.11_{\pm0.57}}$ |

embedding components complement each other and work together to achieve better paper embedding.

*4.4.2 Embedding Analysis.* To deeply analyze the experimental results, we map different features into a 2-dimensional space by t-SNE [21]. The visualization shows in Figure 4, that each point represents one paper, and the points with the same color represent the same author. We choose *hong_fan* in AMiner-18 as an example.

To analyze the closer-together effect of different variants of the papers in the same $C^a$ and push away of negative instance, we analyze the layout of the blue points circled with red elliptical dashes in different representation spaces. First, Figure 4a divides blue points into two groups that are far away from each other, which indicates that the traditional supervised approach using pairwise binary cross-entropy loss function is unable to aggregate $C^a$ with large differences in representation. Second, Figure 4b and Figure 4c bring the two groups of papers in the same class closer together, achieving better clustering results. At the same time, there is still the

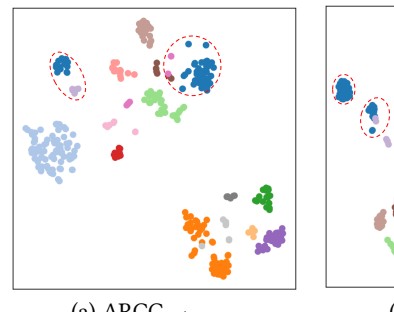

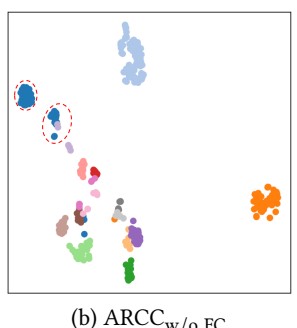

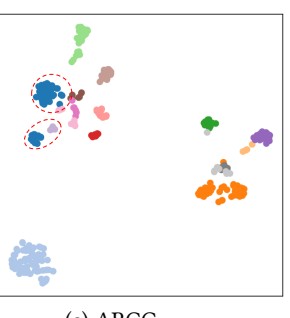

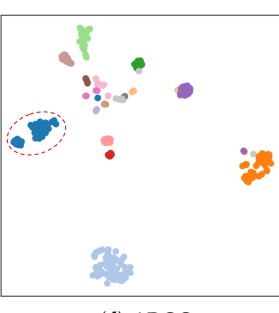

(a) ARCC$_{w/o\ CCL}$
80.47 / 90.10 / 85.01

(b) ARCC$_{w/o\ FC}$
86.34 / 92.23 / 89.19

(c) ARCC$_{w/o\ CC}$
91.72 / 91.70 / 91.71

(d) ARCC
96.06 / 96.86 / 96.45

**Figure 4: The t-SNE visualization of name *hong_fan* in AMiner-18. The dots with the same color represent papers of the same author. (a), (b), and (c) respectively denote the results of different variants. The number in brackets is (Pre / Rec / F1)**

**Table 6: Results for different features.**

| AMiner-18 | | | |
|---|---|---|---|
| **Model** | **Pre** | **Rec** | **F1** |
| Sem | $69.69_{\pm0.57}$ | $72.91_{\pm0.61}$ | $69.41_{\pm0.49}$ |
| Rel | $\mathbf{79.39_{\pm0.36}}$ | $77.22_{\pm0.65}$ | $76.93_{\pm0.44}$ |
| GCN-Fusion | $69.38_{\pm0.32}$ | $73.75_{\pm0.19}$ | $69.69_{\pm0.30}$ |
| ARCC | $78.09_{\pm1.25}$ | $\mathbf{82.32_{\pm0.70}}$ | $\mathbf{78.96_{\pm0.53}}$ |
| **WhoisWho-SND** | | | |
| Sem | $69.82_{\pm0.64}$ | $78.74_{\pm0.24}$ | $72.22_{\pm0.47}$ |
| Rel | $74.59_{\pm0.96}$ | $62.66_{\pm0.78}$ | $66.02_{\pm0.87}$ |
| GCN-Fusion | $74.41_{\pm0.66}$ | $82.51_{\pm0.56}$ | $76.72_{\pm0.49}$ |
| ARCC | $\mathbf{79.52_{\pm0.74}}$ | $\mathbf{83.15_{\pm0.48}}$ | $\mathbf{80.11_{\pm0.57}}$ |

problem of being indistinguishable from other papers in different classes. This suggests that it is difficult to obtain more discriminative representations by ignoring the informational importance of the different features. Finally, Figure 4d achieves the best clustering performance, bringing papers of the same class closer together and pushing away papers of other classes. Our model simultaneously optimizes the representations within the respective features and also combines the complementary disambiguation features contained in the different features. Overall, Figure 4d has a clearer boundary than others, indicating that our model obtains discriminatively more robust representations, which improves the global disambiguation performance.

Based on the above analysis, we believe that the compositional embedding is learned to be more discriminative and brings more generalized features to improve the final paper cluster performance.

*4.4.3 Contributions of Different Features.* To evaluate the performance of the structure and semantic feature, we also compare our model performance at different variants:

**Sem** only uses the semantic embedding produced by the semantic encoder for disambiguation. The semantic feature includes title, abstract, and keyword information.

**Rel** only uses the relation embedding produced by the relation encoder. The relation feature contains co-occurrence relations including co-author, co-institution, and co-venue.

**GCN-Fusion** incorporates semantic and relation features using a traditional GCN model.

Table 6 shows the results of different variant models using different features. It can be observed that single-feature representation can not effectively solve the disambiguation problem. In comparison with our proposed ARCC framework, the performance of the Sem decreases by 9% and of the Rel decreases by 2% on AMiner-18, and the performance of the Sem decreases by 7% and of the Rel decreases by 14% on WhoisWho-SND. The results show the limitations of a single feature.

In addition, the performance of the GCN-Fusion using traditional GCN combining semantic and relational features decreases by 9.2% and 3.3% compared to our proposed ARCC framework on AMiner-18 and WhoisWho-SND, respectively. The performance of the GCN-Fusion is even lower than the Rel on AMiner-18. The possible reason for the result is that mixing the semantic features with other papers via GCN models risks contaminating the semantic feature of the document of interest. This reveals that using the GCN approach to fuse semantic and relational features may be disadvantaged in the author name disambiguation task.

## 5 CONCLUSION

This paper addresses the challenges posed by noisy paper association graphs and inadequate learning algorithms for representation learning in the context of author name disambiguation. To tackle these issues, we propose an iterative paper association refinement process that successively utilizes a rule-guided graph refinement approach and a semantic-guided graph refinement approach to reduce false edges and recover missing edges. Additionally, to enhance the discriminative power of learned representations, we present a compositional contrastive embedding approach that performs contrastive learning on individual semantic embedding and structure embedding, and their fused embeddings. Through empirical evaluations on two real-world datasets, we demonstrate that our proposed method, ARCC, surpasses existing state-of-the-art models. In future work, we plan to explore and incorporate more principled methods to further refine paper association graphs.

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

# A  DETAILS OF RULE-GUIDED GRAPH

Three judgmental guidelines of $\mathcal{H}$. Specific explanations are as follows:

**The Frequency of Collaboration of Co-authors**: If $P_i$ and $P_j$ have at least two of the same authors(excluding the disambiguation author name, the same as below), the edge of the two papers should be reserved. This is based on the assumption that authors working with multiple identical co-authors are highly likely to be the same person.

**The Intuition of Research Team from Same Institution**: If $P_i$ and $P_j$ share other authors and have at least one common institution, the edge should be selected. This guideline follows such intuition that the author tends to cooperate with other authors in the same institution.

**The Similarity of the Authors' Research Topics**: If the sum of inverse document frequency (IDF) of share words of $P_i$ and $P_j$ is greater than a predefined threshold $t^4$, the edge of two papers should be connected. It holds with the intuition that one author tends to focus on some research topics and then write papers with the same technical terms or special words.

# B  EVALUATION METRICS

To validate the performance of various models, pairwise Precision, pairwise Recall, and pairwise F1-score are widely used in AND tasks. The formulations of these evaluation metrics are as follows.

$$\mathrm{Pre} = \frac{\#PC}{\#TPP} \tag{16}$$

---

[4]We empirically set the threshold as 8.

Table 7: The detailed results on WhoisWho-SND

| name | ARCC | | | MFAND | | | Aminer | | | Beard | | | AGAND | | |
|------|------|------|------|-------|------|------|--------|------|------|-------|------|------|-------|------|------|
| | Pre | Rec | F1 | Pre | Rec | F1 | Pre | Rec | F1 | Pre | Rec | F1 | Pre | Rec | F1 |
| baohong_zhang | 85.76 | 99.92 | **92.30** | 96.54 | 86.53 | 91.26 | 93.18 | 73.27 | 82.04 | 90.33 | 70.56 | 79.23 | 73.00 | 46.27 | 56.63 |
| aiqin_wang | 98.16 | 96.55 | **97.35** | 99.90 | 88.25 | 93.71 | 90.69 | 99.85 | 95.04 | 87.60 | 77.25 | 82.10 | 91.12 | 50.00 | 64.56 |
| haibo_he | 99.39 | 93.18 | **96.18** | 99.99 | 88.14 | 93.69 | 97.92 | 38.53 | 55.30 | 95.24 | 50.00 | 65.57 | 94.78 | 28.33 | 43.62 |
| bing_ren | 96.85 | 97.54 | **97.19** | 91.81 | 72.09 | 80.76 | 90.91 | 97.17 | 93.94 | 86.55 | 75.11 | 80.42 | 87.14 | 26.38 | 40.49 |
| jijun_zhao | 86.02 | 91.86 | 88.84 | 98.89 | 84.29 | 91.01 | 94.46 | 95.69 | **95.07** | 91.25 | 84.26 | 87.61 | 72.08 | 49.78 | 58.88 |
| frank_caruso | 81.63 | 71.50 | **76.23** | 79.94 | 60.67 | 68.98 | 76.71 | 36.25 | 49.23 | 70.57 | 40.03 | 51.08 | 95.76 | 25.00 | 39.64 |
| xiaohong_guan | 98.28 | 96.59 | **97.43** | 88.02 | 91.51 | 89.73 | 78.16 | 55.19 | 64.70 | 77.36 | 56.12 | 65.05 | 75.33 | 20.13 | 31.77 |
| david_parker | 64.90 | 98.92 | **78.38** | 67.11 | 84.28 | 74.72 | 56.50 | 55.19 | 71.49 | 54.26 | 83.63 | 65.81 | 85.65 | 36.10 | 50.79 |
| hongjun_song | 84.54 | 88.44 | 86.45 | 99.70 | 78.58 | 87.89 | 99.58 | 95.19 | **97.34** | 89.33 | 82.13 | 85.57 | 73.91 | 46.87 | 57.36 |
| min_hu | 84.10 | 81.95 | **83.01** | 77.54 | 74.94 | 76.22 | 81.27 | 63.93 | 71.57 | 82.30 | 52.61 | 64.18 | 90.23 | 22.06 | 35.45 |
| jie_tang | 94.98 | 94.90 | **94.94** | 90.31 | 77.55 | 83.44 | 71.27 | 29.69 | 41.91 | 73.25 | 35.40 | 47.74 | 79.70 | 23.42 | 36.20 |
| feng_wang | 73.71 | 90.15 | **81.11** | 48.02 | 87.22 | 61.94 | 59.71 | 66.67 | 63.00 | 54.33 | 65.33 | 59.32 | 89.46 | 25.32 | 39.46 |
| jian_pei | 96.04 | 96.46 | **96.25** | 59.81 | 95.56 | 73.57 | 96.07 | 65.61 | 77.97 | 90.16 | 63.19 | 74.30 | 73.68 | 35.11 | 47.55 |
| haining_wang | 83.28 | 97.97 | **90.03** | 54.18 | 81.95 | 65.23 | 85.82 | 43.13 | 57.41 | 83.42 | 70.66 | 76.51 | 85.83 | 52.75 | 65.34 |
| r_gupta | 90.41 | 86.86 | **88.60** | 83.76 | 58.78 | 69.08 | 92.48 | 79.17 | 85.31 | 87.71 | 75.33 | 81.05 | 82.31 | 48.77 | 61.24 |

$$\text{Rec} = \frac{\#PC}{\#TP} \tag{17}$$

$$\text{F1} = \frac{2 \times \text{Pre} \times \text{Rec}}{\text{Pre} + \text{Rec}} \tag{18}$$

where $\#PC$ means the number of pairs correctly predicted to the same author, and $\#TPP$ means the number of total pairs predicted to the same author, and $\#TP$ means the number of total pairs to the same author.

## C DETAILS RESULTS ON WHOISWHO-SND

To detailed analyze overall results on WhoisWho-SND. Table 7 shows the results of 15 name references sampling from the test dataset. It can be observed that ARCC outperforms other baselines in 13 name references. These results confirm the effectiveness of our proposed ARCC model.

