# OpenReview forum: "Author Name Disambiguation via Paper Association Refinement and Compositional Contrastive Embedding"
_ACM.org/TheWebConf/2024/Conference — TheWebConf24_

### Official Review · Reviewer_aTKR · 2023-11-21

**Novelty:** 4
**Technical Quality:** 4

**Review:**

This paper proposes a novel association refining and compositional contrasting (ARCC) framework to address the author name disambiguation task. The idea makes sense, and the paper is easy to follow. Below are the pros and cons:

Pros:
1. The studied problem is significant.
2. The observation of uncertainty in paper association graphs makes sense.

Cons:
1. There are some typos in the paper. For example,
(1) in line 88, textural -> textual.
(2) in line 226, Supcon -> SupCon.
(3) in line 230, infromation -> information.
(4) in line 493, lacking space before "of".
(5) in line 576, Amier -> AMiner.
2. The authors need to compare their methods with more baselines (see below).
3. Some specifics for experiments are lacking (see below).

**Questions:**

1. Since WhoIsWho competitions open-sourced winning solutions, the authors need to compare their method with more baselines, such as SND-all in [1]. Authors can submit their results on WhoIsWho leaderboard to demonstrate the effectiveness of their method.
2. There seems to be no ablation study when making paper association graphs fixed.
3. The detailed methods for building co-author/co-organization/co-venue edges are lacking. The detailed definitions of paper semantic embeddings are lacking.
4. Why are not there negative samples in Eq (14)?
5. Why is Recall more important? If regarding all papers in one cluster, Recall is 1, but the disambiguation results are bad.

[1] Chen, Bo, Jing Zhang, Fanjin Zhang, Tianyi Han, Yuqing Cheng, Xiaoyan Li, Yuxiao Dong, and Jie Tang. "Web-Scale Academic Name Disambiguation: the WhoIsWho Benchmark, Leaderboard, and Toolkit." KDD 2023. https://github.com/thudm/whoiswho

**Reviewer Confidence:**

4: The reviewer is certain that the evaluation is correct and very familiar with the relevant literature

**Scope:**

4: The work is relevant to the Web and to the track, and is of broad interest to the community

---

### Official Review · Reviewer_u4EB · 2023-11-23

**Novelty:** 6
**Technical Quality:** 6

**Review:**

In this paper, the authors propose their methodology for addressing one of the most crucial task in the context of bibliographic databases, i.e. the AND (author name disambiguation) task. They tested their approach against existing datasets and methodologies (using the results in the related papers).

The rationale of the work is sound, and it is written appropriately and clearly. The experiments introduced are against the state of the art and are appropriate.

The main issue is that some important existing works have not been considered either in the section describing related works and also in the experimentation. For instance, the following articles described different approaches for AND tasks, and they have used AMiner to experiment with them:

1. Fan, X., Wang, J., Pu, X., et al. (2011). On graph-based name disambiguation. Journal of Data and Information Quality, 2(2), 1–23. https://doi.org/10.1145/1891879.1891883

2. Santini, C., Gesese, G.A., Peroni, S. et al. A knowledge graph embeddings based approach for author name disambiguation using literals. Scientometrics 127, 4887–4912 (2022). https://doi.org/10.1007/s11192-022-04426-2

3. Chen, Y., Yuan, H., Liu, T., et al. (2021). Name disambiguation based on graph convolutional network. Scientific Programming, 2021, e5577. https://doi.org/10.1155/2021/5577692

In addition, other relevant works have been recently published which address similar topics, proposing new approaches or reviewing existing ones:

Gong, J., Fang, X., Peng, J. et al. MORE: Toward Improving Author Name Disambiguation in Academic Knowledge Graphs. Int. J. Mach. Learn. & Cyber. (2022). https://doi.org/10.1007/s13042-022-01686-5

Michele De Bonis, Fabrizio Falchi, Paolo Manghi:
Graph-based methods for Author Name Disambiguation: a survey. PeerJ Comput. Sci. 9: e1536 (2023) https://doi.org/10.7717/peerj-cs.1536

Including and discussing all of these is a crucial aspect to consider.

In section 4, the datasets used should be detailed a bit better, in particular in the content of the dataset. For instance, it is not enough to list how many papers and author names are there, but also the provenance of such papers and authors. Approaches for AND tasks may vary greatly depending on the authors' population, e.g., if the authors are mainly from European countries or Asian countries. For instance, the number of homonymous names in South Korea is more significant than that of possible homonymous names in Germany.

Silvio Peroni
https://orcid.org/0000-0003-0530-4305

**Questions:**

1. The approach proposed has been experimented with a dataset where all the authors of each paper are seen as separate entities (at least before the running of the clustering methodology). Did the authors try to assess their approach when some of the authors have been already disambiguated - thus, in the input datasets, some of the people behind the names have been already identified, for instance, because they have a disambiguating ID (e.g. an ORCID) assigned? It would be essential since several existing bibliographic databases have a mixed scenario of authors that have been already disambiguated (thanks to persistent identifiers) and others that practically are described only via their names.

2. Did the authors test their approach by also trying to use, as input, an actual bibliographic database instead of a benchmark like AMiner?

**Reviewer Confidence:**

4: The reviewer is certain that the evaluation is correct and very familiar with the relevant literature

**Scope:**

4: The work is relevant to the Web and to the track, and is of broad interest to the community

---

### Official Review · Reviewer_oJX4 · 2023-11-24

**Novelty:** 4
**Technical Quality:** 6

**Review:**

Strengths:
S1. The paper is well-written and the idea is motivated.

S2. The related work is sufficient but can be improved with better explanation of how the existing work is different from this work.

S3. The proposed method outperformed baselines.

S4. The authors conduct several ablation studies to analyze the effectiveness of components in the method.

Weaknesses:
W1. The number of authors is needed to run their method.

W2. Eliminating uncertainties in the paper association graph is the key idea of this work, leading to the proposed paper association refinement approach. However, It is unclear how big the impact of uncertainties in the paper association graph and how much uncertainties after the refinement process compared to graph-based baselines. Figure 3b in Section 4.3.2, if correct, shows a mixed story: as they continue the training, the number of true edges decreases and the number of increased false edges seems stable (~100) — displaying a trend that the graph becomes more uncertain after each iteration. This suggests that additional analysis may be needed.

**Questions:**

Q1. What optimization technique do you use to enforce the constraint in Eq 12?

Q2. In the semantic-guided graph refinement in Section 3.4.2, can you clarify about (t) in Eq 5? During training, do you update  A for every training step, or do you run the refinement process for N iterations for each training step and what is the stopping criterion? During testing, T_ae and T_de functions do not have (t) in their signatures, does it mean you only need to run it once?

Q3. In Section 3.4.2, you mentioned that during training the node embeddings X are updated as well. Can you elaborate on that?

**Reviewer Confidence:**

3: The reviewer is confident but not certain that the evaluation is correct

**Scope:**

3: The work is somewhat relevant to the Web and to the track, and is of narrow interest to a sub-community

---

### Official Review · Reviewer_VWjH · 2023-11-25

**Novelty:** 5
**Technical Quality:** 5

**Review:**

This paper studies the task of author name disambiguation, which is to cluster papers of the same author name into clusters corresponding to the actual author identity. The proposed method include two major parts: the first part include a set of new rules on constructing paper-paper relations and a iterative graph refinement based on learned paper similarity, and the second part introduces a supervised contrastive loss for learning both semantic and structure information of paper graph.

The proposed method shows strong performance compared with baselines. The newly introduced rules on graph feature refinement seems to bring advancement to the task and the learning method also looks effective. The authors provide a wide range of comparative analysis to show the effectiveness of different parts of the new method.

For weaknesses, first, I would like to see more analysis on the newly proposed heuristics for initial graph refinement. Although the newly added rules make sense to me, it is still meaningful to see how the method performs if only removing those new rules (I didn't find such ab ablation) and compare with baselines. Besides, some analysis can be helpful to show the rules are not ad hoc on the data. Second, the number of hyperparameters are quite large, which can makes it costly to do grid search and diminish the applicability. Some parameter sensitivity analysis should be helpful.

Overall, the paper is clearly written and deliver a novel method for the author disambiguation task.

**Questions:**

- How is the constraint in equation 12 imposed in the training process? I wonder why the authors does not use softmax for normalization
- For the task setting, I wonder in practice where does the number of clusters, K, come from, and if the number K is already known, then there should also be some supervision available for this author name and thus it does not need to be formulated as a clustering task?
- I am curious that why pairwise is used in the task given its clustering nature. Why not using cluster alignment based evaluation?

**Reviewer Confidence:**

3: The reviewer is confident but not certain that the evaluation is correct

**Scope:**

3: The work is somewhat relevant to the Web and to the track, and is of narrow interest to a sub-community

---

### Official Review · Reviewer_sjkQ · 2023-11-26

**Novelty:** 5
**Technical Quality:** 5

**Review:**

This paper presents the ARCC system for author name disambiguation. Instead of simply using textual features or associations (e.g., co-author, co-institution, etc.), the authors proposed a two-stage process: association refinement to reduce uncertainties and also a contrastive learning approach to learn better representations. The authors evaluated the proposed ARCC system on two benchmark datasets and demonstrated noticeably better F1 scores on both. The authors also carried out an ablation study to show the effectiveness of the different components. Generally speaking, I think this is an interesting study with good results.

Pros:
1) An interesting refinement idea on the associations (e.g., edge dropping and adding).

2) An effective contrastive learning idea without which the system would have a substantial performance drop (Table 5).

3) Good comparison to several SOTA systems.

**Questions:**

I have a couple of clarification questions on the experimental setup and results:

1) In Table 4, I can see ARCC generally performs better than other systems on different author names. However, it seems to perform particuarly worse than others for this name "yanqing_wang". To me, this name doesn't seem to be very ambiguous and there are certainly more difficult ones (e.g., rong_yu). Any insights on this observation?

2) I am curious about the trade-off between precision and recall. In Table 2, ARCC has the highest F1 score but at the same time, its precision seems to be lower than some other systems. Although we want high recall, precision is also important for linking different authors' publications. Was the "edge adding" a bit more aggressive on this dataset?

3) A couple of minor questions: How was the hidden dimension of 100 determined and any early stopping used during training?

**Reviewer Confidence:**

3: The reviewer is confident but not certain that the evaluation is correct

**Scope:**

3: The work is somewhat relevant to the Web and to the track, and is of narrow interest to a sub-community

---

### Decision · Program_Chairs · 2024-01-22

**Decision:**

Accept

**Comment:**

The paper is well written, its problem is clearly relevant and the proposed approach has been appropriately evaluated.
 The thorough discussion has brought some clarification for most of the points raised by reviewers, who all agree that the technical quality is good. Of course if the paper is accepted, then a lot of the provided responses/clarifications (including the one on ablation study, where two reviewers failed to see it in the first place) should be included in the final version, in the main text or as supplementary material.

 PS: note that personally I was not convinced with the point about recall in the response. Having results that mimic the general structures of real bibliographic datasets (including the presence of big clusters) is indeed interesting. But if these big clusters are produced based on wrong author disambiguation, I don't see the point for any real application. This is not a reason to reject the paper, but I would prefer if the authors would weigh this before including their argument on recall in the paper, if it is accepted.